# Advances and Challenges in Cytomegalovirus Detection Methods for Liver Transplant Donors

**DOI:** 10.3390/diagnostics13213310

**Published:** 2023-10-26

**Authors:** Xiaoping Li, Yiwu Zhong, Yinbiao Qiao, Haoyu Li, Xu Hu, Saber Imani, Shusen Zheng, Jianhui Li

**Affiliations:** 1Key Laboratory of Pollution Exposure and Health Intervention of Zhejiang Province, Shulan International Medical College, Zhejiang Shuren University, Hangzhou 310015, China; 2Zhejiang Chinese Medical University, Hangzhou 310053, China; 3Department of Hepatobiliary and Pancreatic Surgery, Department of Liver Transplantation, Shulan (Hangzhou) Hospital, Zhejiang Shuren University School of Medicine, Hangzhou 310022, China; 4NHC Key Laboratory of Combined Multi-Organ Transplantation, Hangzhou 310003, China; 5Jinan Microecological Biomedicine Shandong Laboratory, Jinan 250117, China; 6Shulan International Medical College, Zhejiang Shuren University, Hangzhou 310015, China; 7The Organ Repair and Regeneration Medicine Institute of Hangzhou, Hangzhou 310003, China

**Keywords:** liver transplantation, cytomegalovirus, infection, detection methods, CRISPR–Cas

## Abstract

Cytomegalovirus (CMV) infection is a highly prevalent opportunistic infection among liver transplant recipients. When the liver donor is infected with CMV, there is a risk of transmission to the recipient, leading to CMV infection. To improve the postoperative outcome of liver transplantation, it is crucial to shift the focus of CMV detection to the donor and achieve early diagnosis, as well as implement effective preventative and therapeutic measures. However, the commonly used CMV detection methods in the past had limitations that prevented their early and accurate diagnosis in liver transplant donors. This review focuses on the latest advancements in CMV detection methods that can potentially be applied to liver transplant donors. The objective is to compare and evaluate their clinical utility, thereby providing guidance and support for rapid and accurate diagnosis of CMV infection in the clinic. The clustered regularly interspaced short palindromic repeats-associated proteins (CRISPR–Cas) system-based assay emerges as a promising method for detecting the virus, offering great prospects for early and expedient CMV infection diagnosis in clinical settings.

## 1. Introduction

Cytomegalovirus (CMV) is one of the most opportunistic viruses to which liver transplant recipients are susceptible. CMV infection in liver transplant donors can lead to CMV carriage in the donor’s liver, posing a risk of CMV infection in the recipient. This not only affects the outcome of liver transplantation but also impacts the prognosis and management of the recipients. In severe cases, it can result in the loss of organ function or even endanger the recipients’ lives. While there are various testing methods to determine CMV infection in recipients after liver transplantation, there is a growing trend in the morbidity and mortality rates associated with CMV infection. Therefore, shifting the focus to detecting CMV in the liver transplant donor and implementing effective preventive and therapeutic measures is crucial in improving the postoperative outcome of liver transplantation. This approach holds significant significance. Commonly used methods in the past to detect CMV in liver transplant donors include viral culture, histopathologic examination, antigenemia assay, serologic testing, and the nucleic acid amplification test (NAAT) [1]. However, these methods have limitations and are not sufficient for early and accurate diagnosis of CMV infection. Hence, this paper primarily summarizes the most recent research progress in CMV detection methods that may be applied to liver transplant donors. It also compares and evaluates their clinical utility, aiming to provide a valuable reference for rapid and accurate diagnosis of CMV infection in the clinic.

## 2. CMV Infection and Diagnosis in Liver Transplant Donors

CMV is the most common opportunistic infection in liver transplant recipients, and it has a direct impact on graft survival and patient mortality [2]. The histopathologic features of CMV were first described in 1904, and in 1957, Craig successfully isolated and identified the virus [3]. As a β-herpesvirus, CMV can remain in the host for extended periods and is a significant pathogen in immunocompromised populations [4]. Mature CMV viral particles are enveloped double-stranded linear DNA viruses with a diameter of 150~200 nm, a genome length of 225~240 kb, and encode more than 200 proteins [5]. CMV infection is highly prevalent in the population, and although CMV infection elicits specific humoral and cellular immunity, CMV can hyper- and persistently infect the host [6]. CMV infection is a major cause of morbidity and mortality in organ transplant recipients, even though it is generally clinically insignificant when the host has a normal immune response.

CMV infection is defined as the presence of CMV replication, indicated by the isolation or detection of viral proteins (antigens) or nucleic acids in any body fluid or tissue specimen, regardless of symptoms [7]. CMV disease can manifest in two forms: CMV syndrome and tissue-invasive end-organ disease. CMV syndrome is characterized by the presence of at least two symptoms or signs after CMV infection. These include an unexplained fever lasting at least 2 days, as well as systemic symptoms like malaise, muscle aches, leukopenia, or thrombocytopenia. On the other hand, tissue-invasive end-organ disease is diagnosed through biopsy confirmation and includes conditions such as CMV-associated hepatitis, pneumonia, retinitis, or gastroenteritis [8,9]. Apart from the direct effects of CMV infection and disease, it is essential to consider the “indirect effects”. These effects can be categorized as general or graft-specific and are associated with increased rates of infection, graft loss, morbidity, and mortality across all types of effects [10,11]. The pathogenesis of CMV infection is highly intricate, involving numerous interactions between CMV and the human immune system. These interactions are mediated through various complex mechanisms, including cytomegalovirus effects on human leukocyte antigen expression, cytokine production, and adhesion molecule production [5]. CMV replication can be found in various body tissues, blood, and other fluids, irrespective of symptoms. It can be determined by antigen detection, nucleic acid testing, or viral culture. Depending on the method or principle employed, CMV replication in the blood is referred to as CMV antigenemia (antigen detection method), DNAemia or RNAemia (nucleic acid detection), or CMV viremia (virus culture method) [8].

Table 1 shows the methods commonly used for CMV testing in the past, which were compared and evaluated. Among them, quantitative NAAT-based CMV viral load testing has become a major tool for diagnosing active disease, initiating preemptive therapy, monitoring response to antiviral therapy, and signaling the risk of clinical relapse or antiviral resistance [12]. It has been shown that prophylaxis and preemptive antiviral therapy are the two main strategies for preventing CMV disease [13]. CMV prophylaxis refers to the use of anti-CMV drugs in patients at increased risk of CMV reactivation, while preemptive therapy refers to the administration of anti-CMV drugs only when a patient shows evidence of CMV replication. Studies have shown that both of these strategies can reduce the incidence of CMV disease in solid organ transplants [14], but both require early and accurate diagnosis, so further research to obtain a more rapid and accurate diagnosis of CMV infection is critical [15].

## 3. Methods Commonly Used in the Past for the Detection of CMV

### 3.1. Virus Culture

It can be divided into traditional tissue tube cultures (empty spot assay) and vial cultures, both of which are highly specific for the diagnosis of CMV infection [1]. Compared to traditional culture, vial culture is a rapid culture method that is based on low-speed centrifugation, which allows the detection of early CMV antigens before characteristic cytopathic effects appear in tissue culture, achieving shorter turnaround times (48 h) [21]. In clinical virology laboratories, vial cultures have largely replaced traditional culture methods precisely because of the short time required to obtain results. Nevertheless, the sensitivity of vial cultures is not sufficient for the application and remains significantly lower than that of antigen and molecular assays [22,23]. Additionally, vial cultures share similar limitations in terms of specificity with conventional culture techniques.

### 3.2. Histopathology

The gold standard for diagnosing tissue-invasive CMV disease is a histologic examination of biopsy samples [17]. This method involves identifying enlarged cells and nuclei and detecting the presence of intracellular viral inclusions, usually basophilic intranuclear inclusions (called megaloblasts), although eosinophilic cytoplasmic inclusions may also be observed [16,17] (Figure 1). The diagnosis of CMV infection in tissue sections can also be established by immunohistochemical staining or in situ hybridization methods [16]. However, due to the invasive nature of the procedures involved in histopathology, routine CMV testing of liver transplant donors is not recommended.

### 3.3. Serology

CMV comprises a diverse range of serotypes, each defined by its unique CMV glycoprotein composition. Notably, disparities in viral serotypes between the donor and recipient have been identified as a contributing factor to elevated rejection risks in transplantation settings. Understanding these serotypic variations is crucial for optimizing transplant outcomes [24]. CMV infection initially triggers the production of IgM antibodies, indicating recent or acute infection, followed by the production of IgG antibodies, indicating previous or latent infection. The pre-transplant evaluation of CMV IgG levels in both donors and recipients represents a widely endorsed approach for assessing and stratifying the risk of CMV infection [25]. Through the analysis of CMV IgG results, recipients can be categorized into distinct risk groups, including high-risk (characterized by seropositive donors and seronegative recipients), intermediate-risk (comprising seropositive recipients), and low-risk (encompassing seronegative donors and recipients) categories [26]. This categorization is pivotal in guiding clinical decision-making. It enables healthcare professionals to predict the likelihood of CMV disease occurrence and tailor the use of prophylactic antiviral therapy accordingly. Therefore, the determination of serostatus through serologic testing is a fundamental step in pretransplant assessment, providing a critical foundation for optimizing patient care and outcomes. Research has found that it generally takes 10–14 days after infection to detect CMV-specific IgM antibodies [18], and it takes approximately 2–3 weeks from the onset of symptoms to detect CMV-specific IgG antibodies [27]. Therefore, the detection of CMV antibodies IgM and IgG is not conducive to early diagnosis of CMV infection. Due to the delayed appearance of IgM and the prevalence of IgG, it is easy to be misled by false-negative results, which limits the usefulness of serum antibody testing in the diagnostic process for CMV infection.

### 3.4. Antigenemia

CMV antigenemia is characterized by the detection of CMV antigen, specifically PP65, within peripheral blood leukocytes (PBLs), as illustrated in Figure 2 [8]. This technique employs a semi-quantitative approach to ascertain the presence of the monoclonal antibody PP65 within CMV-infected peripheral blood polymorphonuclear leukocytes (PMN) and employs immunohistochemistry or immunofluorescence methodologies [19]. Under fluorescence microscopy, leukocytes exhibiting positive antigenemia exhibit a uniform yellow–green nuclear pattern. Positive results are quantified as the ratio of stained cells to the total cell count, with diagnostic significance attributed to PP65 antigenemia when one or more positive cells per 200,000 are observed [19]. This method plays a crucial role in detecting ongoing viral replication and aids in clinical diagnosis. Compared to the NAAT assay for CMV, the antigenemia test also showed reliable, rapid, and sensitive results, and both were comparable in diagnosing active CMV infection and guiding treatment response [28]. This assay is sensitive and specific, relatively easy to perform, and does not require expensive equipment, but results are limited by a lack of standardization, including subjective interpretation of results and the need for adequate neutrophil counts (>1000 cells/mL), and importantly, PP65 antigenemia values are usually elevated during the first week of CMV infection; therefore, it is important to evaluate the results of the assay during that period [29,30]. Due to the lack of standardization of antigenemia test results and the requirement for the number of PBLs, most laboratories have moved away from antigenemia testing for CMV testing in liver transplant donors, preferring to use quantitative molecular testing methods.

### 3.5. Cell-Mediated Immunization (CMI) Assay

Multiple previous studies have solidified the connection between CMV-specific T-cell immunity and the presence of CMV viremia, as well as the development of associated diseases [31]. To evaluate CMV-specific cell-mediated immunity (CMI), we conducted an enzyme-linked immunosorbent spot (ELISPOT) assay [32]. This assay involved the stimulation of T cells using overlapping peptide pools derived from CMV phosphoprotein 65 (pp65) and immediate early-1 (IE-1) proteins. Subsequently, we detected interferon-gamma (IFN-γ)-producing cells, primarily originating from CD4+ and CD8+ T cells, using spot-forming units as a measure [32]. This approach provides valuable insights into the functionality of CMV-specific T cells and their potential role in combatting CMV viremia and associated diseases. The CMV–CMI assay has been utilized for the CMV prognosis and prediction [33], as a complement to the CMV viral load monitoring [34,35], and for risk stratification before organ transplantation [36,37]. Some studies have indicated its potential usefulness. Nevertheless, there exist several challenges that need to be addressed for the broad adoption of this approach. These challenges encompass the intricate nature of the experimental procedure, the absence of a clearly defined positive threshold, concerns regarding cost-effectiveness, and the limited availability of clinical trials to establish its efficacy. Furthermore, it has been observed that the detection of CMV-specific cell-mediated immunity (CMI) may be less informative in the case of deceased donors due to the high occurrence of indeterminate results [38]. These complexities underscore the need for further research and optimization to fully realize the potential of CMV-specific CMI assessment in clinical practice.

### 3.6. Quantitative Nucleic Acid Amplification Test (QNAT)

The quantitative polymerase chain reaction (qPCR)-based QNAT is a quantitative assay used to measure CMV viral load in clinical samples by detecting and amplifying small amounts of viral nucleic acid through PCR technology [12]. Due to its high sensitivity and throughput, qPCR has become the preferred diagnostic assay for CMV infection, enabling clinicians to diagnose and treat infections effectively. It is widely utilized in the clinic to guide preemptive therapy, assess the efficacy of antiviral therapy, determine the timing of therapy, and monitor viral replication and disease progression [12,39]. There is a correlation between the quantification of CMV DNA and viral load (i.e., the degree of viral replication), and active CMV replication is manifested by higher viral load values or an increasing trend in viral load [40]. Real-time quantitative PCR assays are commonly used in clinical laboratories to measure CMV viral load. However, variability in the results persists due to differences in sample types, nucleic acid extraction techniques, target genes, primers, probes, detection methods, and quantification standards used in different laboratories [41,42]. Despite the introduction of international standards by the World Health Organization (WHO) that has enabled the standardization of viral load values derived from assays developed by different laboratories [43], which is conducive to improving the consistency of assay results, the problem of variability between different PCR assay platforms or methods still exists [44,45]. In recent years, it has also been demonstrated that the amplicon size and DNA extraction method of CMV qPCR assays affect the variability [46] and that sample type (e.g., plasma versus whole blood versus peripheral blood mononuclear cells) also greatly affects the variability of viral load values [47]. Newer methods, such as Droplet Digital PCR, are becoming more widely used in quantitative viral load assays, not only showing superiority to qPCR in dilution assays of synthetic DNA [48] but also improving consistency between methods while reducing variability in results without relying on quantification standards [49]. However, the use of digital PCR as a reference standard or for routine clinical testing still requires thorough validation of any given assay and instrumentation, and there are currently no commercially available analytical methods that use digital PCR assays. In addition, reverse transcriptase nucleic acid amplification (RT-PCR), which can detect viral mRNA transcripts in peripheral blood leukocytes without relying on the presence of DNA, can help to diagnose active CMV infection, but it is of less sensitivity than the PP65 antigen detection and PCR [50].

## 4. Recent Advances in CMV Detection Methods for Liver Transplant Donors

### 4.1. Nucleic Acid Sequence-Based Amplification (NASBA)

Isothermal amplification stands as a compelling alternative to traditional quantitative polymerase chain reaction (qPCR) techniques, offering the ability to amplify nucleic acids at a constant temperature without the need for costly thermal cyclers [51]. Among these isothermal methods, nucleic acid sequence-based amplification (NASBA) emerges as a specific and robust technology capable of detecting unclipped viral mRNA within a background of DNA [52], as illustrated in Figure 3. NASBA operates optimally at a temperature of 41 °C, allowing for cost-effective and user-friendly heating approaches. Importantly, NASBA exhibits the unique ability to selectively amplify RNA in the presence of coexisting background DNA and DNA target sequences, provided they have undergone prior denaturation [53].

The detection of mRNA PP67, indicative of active viral replication, serves as a valuable marker for CMV infection [16]. One notable example of a specific test harnessed through NASBA technology is the Nuclisens pp67 test, designed to monitor the expression of CMV mRNA PP67. Studies have demonstrated its high specificity and lower sensitivity in comparison to antigenemia and PCR-based tests [54]. While NASBA is widely employed for identifying various pathogenic microorganisms and holds promising research potential, its effectiveness may be constrained by prerequisites for pre-amplification nucleic acid extraction and purification steps [55]. Additionally, its relatively lower sensitivity and susceptibility to RNA mutations, potentially leading to false positives, may limit its utility when compared to antigenemia detection and quantitative nucleic acid testing (QNAT) [21]. As such, further refinements are necessary to fully harness the potential of NASBA for CMV detection in liver transplant donors.

### 4.2. Loop-Mediated Isothermal Amplification (LAMP)

LAMP is a novel gene amplification technique that is characterized by rapidity, simplicity, and specificity [56]. In the LAMP reaction, pairs of internal and external primers are used to generate a large number of DNA amplification products with complementary sequences and alternate repeat structures by sequential repetition of both types of elongation reactions by strand-displacing DNA polymerase in a burst amplification process [57]. In contrast to real-time qPCR, which necessitates programmed temperature increases or decreases, LAMP has a short turnaround time, does not require expensive instruments, and has the potential for use in point-of-care diagnostic tests [51]. The potential of LAMP for point-of-care diagnostic tests is not only beneficial for the early and precise diagnosis of CMV infection in liver transplant donors but also provides greater diagnostic value for early CMV detection in post-transplant recipients. The integration of isothermal amplification methods into microfluidic devices has demonstrated endless opportunities for rapid, simple, and sensitive detection of pathogens [58]. Although LAMP is a rapid and sensitive nucleic acid amplification technique that can be applied to clinical diagnosis and pathogen detection [59], its complex ring primer design and unstable single-base resolution are important factors hindering its further clinical diffusion [60,61]. Therefore, further research is necessary to use LAMP widely for the early diagnosis of CMV infection.

### 4.3. Hybrid Capture Assay

The hybridization capture assay employs an RNA probe in the form of enzyme-linked immunosorbent assay (ELISA) types to detect and quantify viral DNA, utilizing chemiluminescence to measure the resulting signal. This assay demonstrates comparable sensitivity and specificity to the PP65 antigen assay while outperforming the cell culture assay in determining CMV viremia [62]. However, further investigation is necessary to enhance the diagnosis of CMV infection.

### 4.4. Gene Sequencing

Peripheral blood rapid whole genome sequencing (rWGS) has been used to identify microbial DNA during acute infections [63]. rWGS may be a sensitive method for detecting CMV infections and can be performed to guide the initiation of antiviral therapy, helping to improve patient prognosis [64]. Some studies have shown that rWGS may be more sensitive than qPCR [65], but further studies are needed to determine the specificity and sensitivity of this method and, thus, demonstrate its clinical utility. Metagenomic next-generation sequencing (mNGS) technology is a non-invasive assay that allows for unbiased, hypothesis-free identification of a wide range of pathogenic microbial infections for diagnosis and monitoring of infectious diseases [66]. In addition to pathogen identification, mNGS can also detect virulence genes and resistance genes (Dulanto Chiang and Dekker, 2020). It has been shown that mNGS can simultaneously monitor and quantify multiple viruses in patients with results comparable to standard qPCR assays [67]. However, high detection costs, low detection throughput, and a complex process are challenges that must be addressed before mNGS can be widely adopted in clinical settings.

### 4.5. Gene Chip Technology

Gene chip technology, also known as DNA microprobe arrays, uses in situ synthesis or microdot sampling to bind specific oligonucleotide sequences or cDNA fragments to probes immobilized on a support. These probes are then hybridized with fluorescently labeled samples, with resulting signals analyzed for information on gene expression and arrangement [68]. This method is highly sensitive and accurate, as well as has the advantages of rapidity and high throughput, so it has good application prospects for early diagnosis of CMV infection. Nevertheless, the complex and expensive nature of the technology remains a significant challenge that must be addressed before it can become widely used in the early detection of CMV infection.

### 4.6. CRISPR–Cas System

The CRISPR–Cas system is a sensitive biological system capable of rapidly recognizing pathogen-specific nucleic acids [69]. It is arguably one of the most promising research areas in gene editing technology and the diagnosis of pathogens currently being investigated. The basic principle of the CRISPR–Cas system is depicted in Figure 4. CRISPR–Cas locus structures are composed of the Cas gene upstream, the leader sequences, and CRISPR sequences, which encompass repeat sequences and spacer sequences. The DNA of each spacer sequence is not identical but matches the viral DNA [70]. The Cas gene expresses the Cas protein with helicase and nuclease activities, which can cut the DNA strand and play a key role in realizing the function of the CRISPR–Cas system. First, the CRISPR sequences transcribe and process the crRNA, which matches the spacer sequences. Then, the crRNA guides the Cas protein to viral target sequences that match the spacer sequences, and finally, the corresponding structural domain of the Cas protein is targeted to shear the viral target sequences. The corresponding target sequences can be recognized by changing the guiding sequences, which lays a mechanistic foundation for the use of the CRISPR–Cas system for nucleic acid detection [71]. At its core, the CRISPR–Cas system operates as a prokaryotic adaptive immune system, designed to identify and cleave foreign nucleic acids effectively [72]. This intricate system falls into two primary categories: Class 1 and Class 2. Class 1 deploys multi-protein complexes, including type I, type III, and type IV systems, to neutralize foreign nucleic acids. Conversely, Class 2 predominantly relies on a single protein to execute its function, encompassing types II, V, and VI [73]. These versatile CRISPR–Cas systems have found extensive application in the realm of pathogen nucleic acid detection, with a particular emphasis on prominent CRISPR-associated proteins, such as Cas9, Cas12, Cas13, and Cas14 [74]. Among these, Cas12 and Cas13 have garnered widespread use in the detection of both DNA and RNA [75]. These groundbreaking developments underscore the remarkable utility of CRISPR–Cas technology in advancing nucleic acid detection methodologies.

The CRISPR–Cas system may be an ideal method for detecting viruses that meets the requirements of sensitivity, specificity, low cost, speed, ease of use, low equipment requirements, and ease of delivery to the user [76]. This has high application value for early, rapid, and accurate diagnosis of CMV infection in liver transplant donors, guiding preemptive treatment, improving post-transplantation outcomes, and monitoring graft rejection after transplantation. To further enhance its diagnostic capabilities, the CRISPR–Cas system can be combined with isothermal amplification methods such as LAMP, NASBA, and recombinase-aided amplification (RAA) for pre-amplification of nucleic acids [77]. The introduction of isothermal nucleic acid amplification technology can eliminate the reliance on thermal cyclers and allow assays to be performed outside of the laboratory, making point-of-care diagnostic tests possible [78]. Leveraging the unique attributes of Cas proteins, which exhibit precise recognition of target nucleic acids and subsequent activation of cutting activity upon recognition, scientists have pioneered a suite of CRISPR–Cas technologies that are transforming the field of molecular diagnostics. These innovations include the Specific High-sensitivity Enzymatic Reporter unlocking (SHERLOCK) [79], DNA Endonuclease Targeted CRISPR Trans Reporter (DETECTR) [80], and the one-Hour Low-cost Multipurpose highly Efficient System (HOLMES) [77], with their fundamental principles illustrated in Figure 5. Cas effectors, guided by crRNAs, demonstrate extraordinary specificity in recognizing and cleaving template nucleic acids. Upon activation, these Cas effectors exhibit remarkable trans-cleavage capabilities, leading to non-specific cleavage of reporters and generating fluorescence signal readouts.

SHERLOCK and DETECTR diagnostic tools share common attributes, including sensitivity, specificity, cost-effectiveness, and the ability to operate without complex equipment. HOLMES was initially developed by combining PCR with the CRISPR–Cas12a system, requiring a two-step reaction. To streamline this process, HOLMES v2 was created, integrating isothermal amplification technology (LAMP) and the CRISPR–Cas12b system, allowing for one-step detection [81]. Additionally, to reduce reliance on the PAM sequence, researchers used LAMP amplification to design core primers containing the PAM site, enabling the LAMP amplicon to carry a specific PAM site for CRISPR/Cas12a recognition, facilitating the detection of any target sequence [82].

In terms of cost-effectiveness, SHERLOCK technologies are priced at less than $1 per test, and DETECTR and HOLMES have been suggested as economical alternatives, with the potential for further cost reductions through systematic optimization [76]. Therefore, the HOLMES v2 assay platform holds promise for detecting cytomegalovirus genomes in transplantation settings. Despite certain limitations, these assays open novel pathways in pathogen detection and offer significant potential for the early diagnosis of CMV infection in liver transplant donors.

The CRISPR–Cas system, beyond its role as a powerful gene-editing tool, has emerged as a robust diagnostic technology, exemplified by the HOLMES-based assay system, heralding a transformative era in molecular diagnostics. As researchers delve deeper into the capabilities of Cas12, Cas13, and Cas14 in trans-cleavage activity and continue to innovate in nucleic acid detection within the rapidly evolving field of CRISPR–Cas applications, it presents an exciting avenue for exploration. Moreover, biosensing platforms based on CRISPR–Cas systems hold tremendous potential to revolutionize pathogen diagnosis [83,84]. These advancements collectively usher in a new era of precision diagnostics and pathogen detection.

Despite the advantages of high specificity, sensitivity, rapidity, and cost-effectiveness, the practical application of CRISPR–Cas system-based pathogen detection methods faces challenges such as off-target effects, sample cross-contamination, and pathogen quantification problems [85]. Nonetheless, the rapidly developing CRISPR–Cas system-based assay technology shows immense promise and has a wide range of potential applications. As one of the most promising areas of current research, the CRISPR–Cas system has significant prospects for early and rapid detection of CMV infection in clinical settings.

## 5. Discussion

CMV is the predominant opportunistic infection in liver transplant recipients. CMV infection in liver transplant donors can lead to CMV presence in the donor’s liver, posing a risk to the recipient. This, directly and indirectly, influences the morbidity and mortality of transplant recipients. Therefore, early detection of CMV in liver transplant donors and implementation of preventive and therapeutic measures are crucial for enhancing postoperative outcomes. Several methods have been employed to detect CMV. A comparative analysis of the specificity and sensitivity across various CMV assays has yielded noteworthy findings. CMV–CMI, molecular assays, gene chip technology, and CRISPR–Cas system-based assays have demonstrated high levels of both specificity and sensitivity. In contrast, antigenemia-based assays exhibit moderate specificity and lower sensitivity. Serologic assays exhibit high specificity but lower sensitivity, while isothermal nucleic acid amplification techniques demonstrate relatively lower levels of both specificity and sensitivity. It is essential to emphasize that while specificity and sensitivity are critical parameters, the choice of assay should also consider additional factors such as time, cost, required equipment, and laboratory conditions. These factors should be carefully evaluated on a case-by-case basis to make an informed selection. Firstly, virus culture, while common, is time-consuming, labor-intensive, and lacks sufficient sensitivity, leading to its declining clinical use. Histopathology remains the diagnostic gold standard for CMV tissue invasive disease but is invasive and not always recommended. Serological detection can be misleading due to the delayed appearance of antibody IgM and the prevalence of IgG. The antigenemia assay is reliable and significant for early diagnosis, but its lack of result standardization and specific requirements limit its use. CMV–CMI shows promise in viral load detection, but its complexity, cost-effectiveness, and limited clinical trials present challenges. QNAT, with the advantages of high sensitivity, high throughput, and high specificity, is currently the preferred method for early diagnosis of CMV infection and is the most widely used pathogen detection method. Still, there is the problem of variability between different detection platforms or methods, which requires further improvement. Secondly, it is worth noting that more and more laboratories are gradually abandoning single-laboratory tests and switching to commercially available tests calibrated to international standards, and the consistency of results across different testing platforms will be further improved. Thirdly, mNGS is also a widely used method for pathogen detection, but the high cost, low throughput, and complexity of the process are challenges that need to be solved for its clinical promotion, and further research is needed. Fourthly, emerging detection methods, like the CRISPR–Cas system and gene chip technology, have a great application prospect, among which the detection technology based on the CRISPR–Cas system is developing rapidly, with the advantages of high sensitivity and specificity, low cost, rapidity, and ease of use, which may become an ideal method for virus detection in the clinic and soon become a preferred clinical method for virus detection, pending further research and validation.

## 6. Summary and Outlook

CMV is among the opportunistic viruses to which liver transplant recipients are most susceptible, and infection in liver transplant donors can lead to CMV carriage in the donor’s liver, thereby placing the recipient at risk. Shifting CMV detection to the liver transplant donor and implementing effective preventive and therapeutic measures are crucial for reducing both the incidence of CMV infection in liver transplant recipients and the mortality rate from complications such as graft rejection. The commonly used CMV detection methods in the past are not sufficient for early and accurate diagnosis of CMV infection in liver transplant donors due to various limitations. Therefore, it is necessary to develop a new type of detection method that meets the requirements of the clinic. Table 2 presents an overview of the advantages and disadvantages associated with the latest CMV detection methods. Currently, QNAT and mNGS are the predominant pathogen detection methods, with QNAT being favored for CMV detection due to its high sensitivity and specificity. This method is vital for early diagnosis and preemptive treatment guidance, but the variability between different PCR detection platforms must be addressed. Emerging detection technologies like the CRISPR–Cas system and gene chip technology hold promise, with the potential breakthrough of combining gene chip technology with enzyme-linked immunosorbent spot (ELISpot) for early CMV detection. The rapidly developing CRISPR–Cas system-based detection technology offers advantages such as high sensitivity, specificity, low cost, and ease of use, positioning it as a potential ideal method for clinical virus detection. However, challenges and opportunities exist, and future development may include integrating with isothermal nucleic acid amplification technology to visualize the detection results, reducing the probability of off-target effects, solving the problems of sample storage and possible contamination, further exploring the types and uses of Cas proteins, and further realizing the combination with the biosensor. A comprehensive assessment of the advantages, disadvantages, and common principles of each detection method is essential, and the limitations of relying on a single detection method must be overcome, such as by integrating CRISPR–Cas reaction with RAA for early and accurate CMV diagnosis in liver transplant donors. Additionally, the possibility of CMV presence in organ irrigation or preservation fluid should be considered, as it may be another checkpoint for secondary CMV infection in transplant recipients. Risk stratification of infection and the adoption of prophylactic and preemptive antiviral treatment for the donor liver or transplant recipients could significantly improve post-transplantation prognosis and management.

## Figures and Tables

**Figure 1 diagnostics-13-03310-f001:**
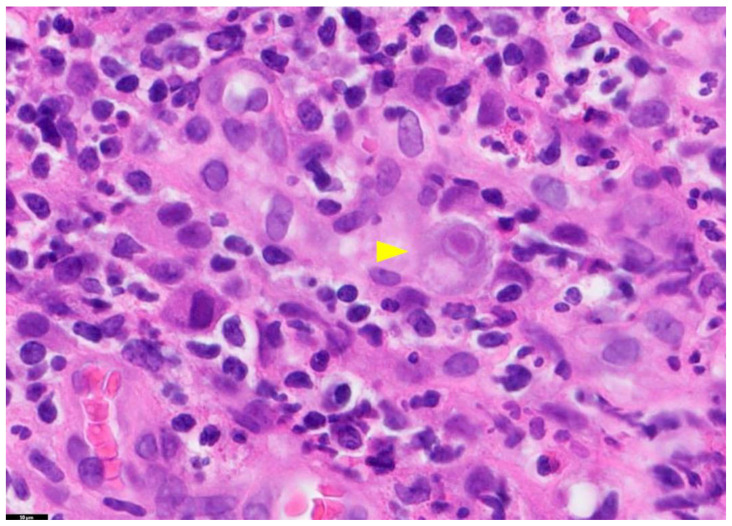
Tissue Section of a CMV-Positive Cell. This figure presents a tissue section of a CMV-positive cell, specifically illustrating CMV gastritis. The yellow arrow in this image highlights a cytomegalic cell with a basophilic intranuclear inclusion encircled by a halo. The staining method used is Hematoxylin and Eosin (H&E), and the magnification is 100×. PathologyOutlines.com, a reputable pathology resource platform, provides the figure. Copyright for this figure is attributed to the original authors.

**Figure 2 diagnostics-13-03310-f002:**
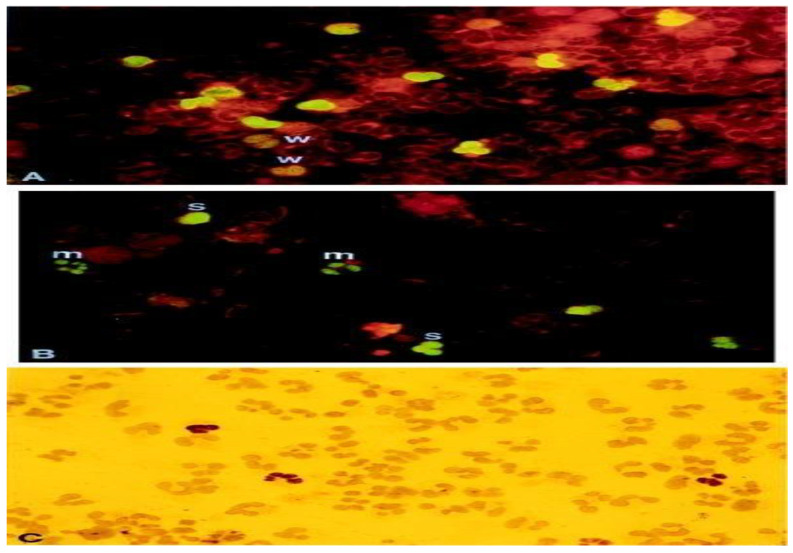
pp65-positive polymorphonuclear (PMN). This figure showcases pp65-positive polymorphonuclear (PMN) cells through different staining techniques. (**A**,**B**) depict indirect immunofluorescence staining, while (**C**) utilizes immunoperoxidase staining. The samples include cytospin preparations of PMN cells containing pp65-positive cells. These samples are obtained from a patient with disseminated HCMV (human cytomegalovirus) infection (**A**) or generated in vitro (**B**,**C**). The degree of staining is denoted as weak (w), moderate (m), or strong (s) based on the intensity of fluorescence. The magnification used for the images is ×910. This figure is sourced from PathologyOutlines.com, a reputable pathology resource platform. Copyright for this figure is attributed to the original authors.

**Figure 3 diagnostics-13-03310-f003:**
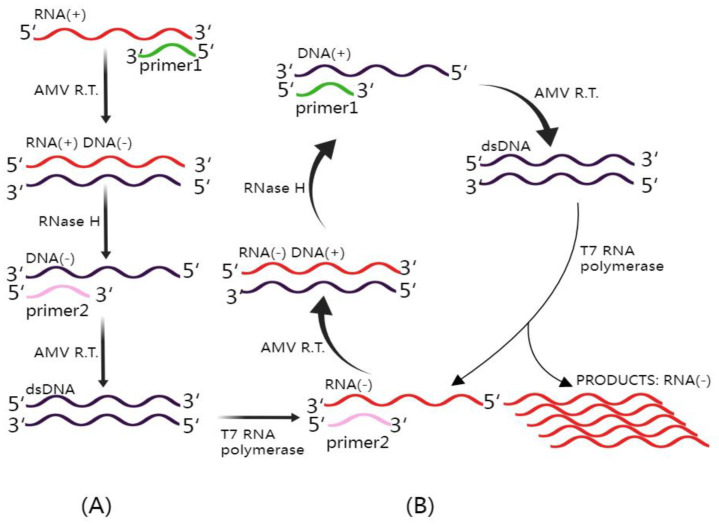
Mechanism of NASBA (Nucleic Acid Sequence-Based Amplification). This figure illustrates the intricate NASBA mechanism. The NASBA reaction mixture includes three essential enzymes: T7 RNA polymerase, RNase H, and avian myeloblastosis virus (AMV) reverse transcriptase, in addition to two specific primers. (**A**) Non-cyclic phase: In this phase, upon the introduction of a target RNA molecule, Primer 1 initiates the reverse transcription process, yielding an RNA–DNA hybrid. Subsequently, the inclusion of RNase H facilitates the degradation of the original RNA within the RNA–DNA hybrids, leaving the complementary DNA (cDNA) available for binding with Primer 2. AMV reverse transcriptase extends the 3′ end of Primer 2, resulting in the formation of double-stranded DNA (dsDNA). This dsDNA is then transcribed by T7 RNA polymerase, ultimately generating antisense RNA. (**B**) Cyclic phase: The antisense RNA proceeds to enter the cyclic phase of NASBA, leading to the amplification of the antisense RNA strand, which is crucial for the sensitive detection of target RNA sequences. This figure was created with Figdraw.

**Figure 4 diagnostics-13-03310-f004:**
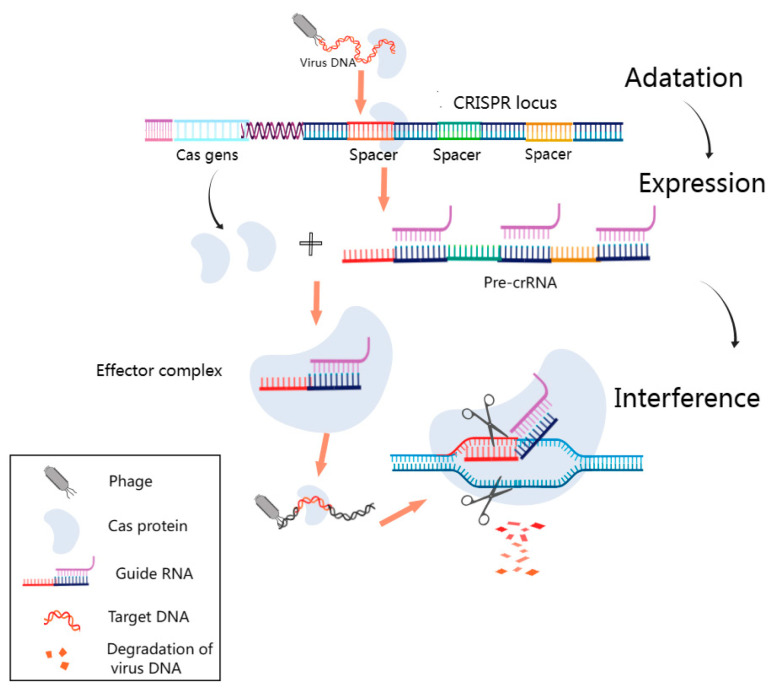
Fundamental Schematic of the CRISPR–Cas System. This figure outlines the pivotal phases in the CRISPR–Cas system’s operation. During the adaptation phase, short DNA fragments resembling viral sequences are incorporated as spacers into designated CRISPR sites. In the subsequent expression stage, the CRISPR sequence is transcribed into precursor CRISPR RNA (pre-crRNA), which is further processed into guide RNA complementing the viral target sequence. Finally, during the interference phase, the guide RNA directs the Cas protein to the virus’s matching spacer target sequence, leading to the formation of an effector complex. The Cas protein’s cleavage activity then degrades the phage’s target DNA. It was created with MedPeer (www.medpeer.cn).

**Figure 5 diagnostics-13-03310-f005:**
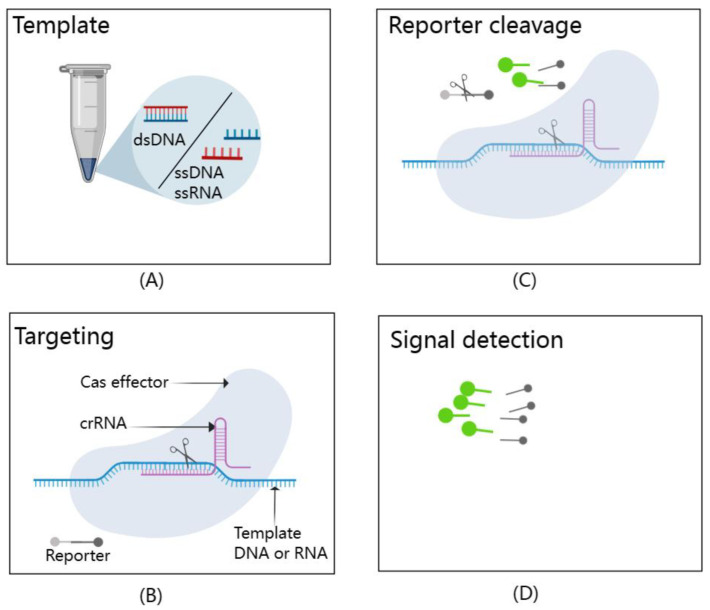
A schematic representation of the CRISPR/Cas-based detection system. It effectively illustrates the dual functionality of Cas effectors in this system. (**A**) Template: The types of targets that can be detected by CRISPR/Cas-based detection systems include dsDNA, ssDNA, and ssRNA. (**B**) Targeting: Cas effectors exhibit exceptional specificity, precisely recognizing and cleaving template nucleic acids with the guidance of crRNAs. (**C**) Reporter cleavage: Once activated, these Cas effectors display impressive trans-cleavage abilities, leading to the non-specific cleavage of reporter molecules. (**D**) Signal detection: This trans-cleavage event generates fluorescence signal readouts, allowing for the detection of target nucleic acids. This figure, created with MedPeer (www.medpeer.cn), provides a concise visual summary of the fundamental principles underpinning CRISPR/Cas-based nucleic acid detection.

**Table 1 diagnostics-13-03310-t001:** Characteristics of commonly used CMV testing methods in the past for liver transplant donors.

Assay	Principle	Advantage	Disadvantage	Reference
Viral culture	Viral replication	Highly specific	Low sensitivity, time-consuming culture, slow turnaround time	[16]
Histopathology	Viral inclusion body	Gold standard for diagnosis of CMV tissue invasive disease; Differentiating CMV disease from allograft rejection	Intrusive operations	[17]
Serology	Specific IgM, IgG antibodies	Liver transplant donor/recipient CMV screening; Predicts the risk of developing disease	Delayed appearance of specific IgM and prevalence of IgG can mislead results	[18]
Antigenemia	PP65 antigen	High sensitivity and specificity; Easy to perform, rapid diagnosis of CMV, no need for expensive equipment	Lack of standardization of results; Some requirement for number of PBLs	[19]
CMV-CMI	IFN-γ produced by CD4+/CD8+ T cells	Commercialized Tests; Prognostic Prediction of CMV; Viral Load Measurement; Pre-transplant risk stratification	Experimental complexity; Lack of positive thresholds; Cost-effectiveness issues; Inadequate clinical trials	[20]
QNAT	Viral load	High sensitivity, high throughput, high specificity	Variability prior to different PCR assay platforms/assays	[12]

Abbreviations: CMV-CMI, Cytomegalovirus-specific cell-mediated immunity; IFN-γ, Interferon-gamma; PBLs, Peripheral blood leukocytes; QNAT: Quantitative nucleic acid testing.

**Table 2 diagnostics-13-03310-t002:** Comparison of the latest CMV detection methods.

Assay	Principle	Advantage	Disadvantage	Specificity	Sensitivity	Refs.
NASBA	mRNA detection	Highly specific for viral replication; clinical utility for preemptive therapy; monitoring response to treatment	Qualitative assay; less sensitive than nucleic acid amplification tests	Low	Low	[21]
LAMP	Single-temperature nucleic acid amplification	Rapid, simple, specific, and not dependent on expensive instruments, with the potential for rapid on-site detection	Complex ring primer design and unstable single-base resolution	Low	Low	[51]
Hybrid capture assay	DNA–RNA hybrid	Highly specific for CMV infection; rapid diagnosis of CMV infection	Less sensitive than nucleic acid amplification tests	High	High	[21]
Gene sequencing	Sequencing the genome	Non-invasive test; provides information on virulence genes and resistance genes; monitors and quantifies multiple viruses in patients	High detection costs; low detection throughput; complex process	High	Medium	[64]
Gene chip technology	In situ synthesis or microdot sampling	High sensitivity and accuracy; fast and high throughput	Costly and technically complex	High	High	[86]
CRISPR–Cas System	Changing the guiding sequences	Meets the requirements of sensitivity, specificity, low cost, speed, ease of use, low equipment requirements, and ease of delivery to the user	Off-target effects; sample cross-contamination; pathogen quantification problems	High	High	[87]

Abbreviations: NASBA, Nucleic Acid Sequence-Based Amplification; LAMP, Loop-Mediated Isothermal Amplification; DNA–RNA, Deoxyribonucleic Acid–Ribonucleic Acid. CMV, Cytomegalovirus; CRISPR–Cas, Clustered Regularly Interspaced Short Palindromic Repeats and CRISPR-Associated.

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
