# Peer review of "Advances and Challenges in Cytomegalovirus Detection Methods for Liver Transplant Donors"

_diagnostics, 2023, doi:10.3390/diagnostics13213310_

Round 1
Reviewer 1 Report
Comments and Suggestions for Authors
Li et al. provided an overview of CMV detection methods in liver transplant donors. They discussed CMV infection and summarized commonly used methods such as viral culture, histopathology, serology, antigenemia, CMV-CMI, and QNAT, comparing their advantages and disadvantages. The authors also outlined the latest CMV detection methods and their pros and cons. The paper aims to facilitate rapid and accurate diagnosis of CMV infection in clinical settings.
To address the revisions:
1. A table should be added to list the pros and cons of the latest CMV detection methods, similar to Table 1.
2. The specificity and sensitivity of both commonly used and latest CMV detection methods need to be compared.
3. More figures should be included to enhance the presentation.
4. Additional references should be included to support the discussed methods and findings more comprehensively.
5. The section on CRISPR-Cas detection methods should be revised to include more details such as the Holmes Cas12a-based detection system and its potential in CMV detection
Comments on the Quality of English Language
Require revisions on English writing.
Author Response
Dear Esteemed Editor and Reviewer #1,
Thank you for your valuable feedback on our manuscript. We appreciate your thoughtful comments and have carefully considered each of your concerns. Here is a brief summary of the changes we have made:
We have addressed the formatting and reference issues, ensuring that individual references are displayed correctly and specialized terms are accurately written.
We have conducted a thorough revision of the entire manuscript, focusing on enhancing its rigor and clarity.
To improve the scientific credibility of the article, we have incorporated relevant new literature.
We hope these revisions meet your expectations and improve the quality of our manuscript.
Thank you for your continued support and guidance.
Best regards,
Yours sincerely,
Xiaoping Li and Jianhui Li
Email: li-xp@zsru.edu.cn
Here is a more detailed response to your specific concerns:
- A table should be added to list the pros and cons of the latest CMV detection methods, similar to Table 1.
Response: Your insightful suggestion for the inclusion of a table listing the advantages and disadvantages of the latest CMV detection methods, similar to Table 1, has been implemented. We have added a new table at the end of our manuscript to provide a comprehensive overview of these methods. Your meticulous review significantly enhances the quality of our work.
- The specificity and sensitivity of both commonly used and latest CMV detection methods need to be compared.
Response: We appreciate your comments regarding the comparison of specificity and sensitivity for both commonly used and the latest CMV detection methods. In response, we have included a comparison in Tables 1 and 2, along with corresponding additions to the manuscript (Line 381~391). Your valuable input has contributed to the comprehensiveness of our research.
- More figures should be included to enhance the presentation.
Response: Thanks to your feedback, we have incorporated three new figures to improve the presentation of our work (Line 115, Line 145, Line 226, Line 338). Your suggestions have enhanced the visual appeal and clarity of our manuscript.
- Additional references should be included to support the discussed methods and findings more comprehensively.
Response: Your suggestion to include additional references to support our discussed methods and findings comprehensively has been addressed. We have added several new references, highlighted in red, to reinforce our research (e.g., 'Li, L.; Li, S.; Wu, N.; Wu, J.; Wang, G.; Zhao, G.; Wang, J. HOLMESv2: A CRISPR-Cas12b-Assisted Platform for Nucleic Acid Detection and DNA Methylation Quantitation. ACS Synth Biol 2019, 8, 2228–2237, doi:10.1021/acssynbio.9b00209').
- The section on CRISPR-Cas detection methods should be revised to include more details such as the Holmes Cas12a-based detection system and its potential in CMV detection.
Response: Thank you for your feedback on the section covering CRISPR-Cas detection methods. We have revised this section to include more details, discussing the HOLMES, SHERLOCK, and DETECTR assay systems and their potential in CMV detection. The CRISPR-Cas system is not only an exceptional gene-editing tool but also a robust diagnostic technology, as exemplified by the HOLMES-based assay system, which is ushering in a transformative era in molecular diagnostics. (Line 309~320, Line 331~366)
Reviewer 2 Report
Comments and Suggestions for Authors
Cytomegalovirus is a ubiquitous pathogen which infect latently in many people. Although the latently infected CMV does not develop obvious disease to healthy individuals, the immune suppression leads to several diseases. The transplant recipients are developed immune suppression by the treatment of immune suppressive drugs such as tacrolimus, therefore the detection of allograft containing CMV is important to understand the prognosis of patients. The review article, diagnostics-2624650 presented the detection methods of CMV from liver transplant donor. This article is well described, but the details of methodology are insufficient. Furthermore, the addition of photographs of histopathology and antigenemia are helpful to understand.
Major points
3.2 Histopathology
It will be better for readers if the photographs of CMV positive cells such as megaloblasts and cytoplasmic inclusions to identify the infected cells.
3.3 serology
It is well known that CMV has several serotypes which determined by the type of CMV glycoproteins. Furthermore, the mismatch of viral serotypes between donor and recipient leads to the increase of rejection. It is necessary to describe the meaning of determination of donor serology and response when the serotypes are different in liver transplantation.
3.4 antigenemia
As described above, it is helpful to insert the photograph of antigenemia positive cells. Furthermore, it is better to describe the criteria of commonly used for antigenemia.
3.5 Cell-Mediated Immunization (CMV) assay
This section describes the ELISPOT or Quantiferon assay. Thess assays are commonly used to measure T cell response against CMV. Therefore, I think this assay is not suitable to detect CMV in donor. Furthermore, which peptides are better to detect CMV-specific T cell response? Authors should describe the meaning to detect anti-CMV profile of donor derived T cell.
4.6 CRISPR-Cas system
The principles of CRISPR-Cas system are described in lines 266-280. However, the methodology of how to detect viral genome is insufficient. Several detection methods (Luc, FISH, etc.) for CRISPR-Cas system are known. Which method is better to detect CMV genome? Authors should discuss the benefits and costs of this system by the comparison of CMV and other viruses which detected by CRISPR-Cas system. In addition, all papers cited in this section is review articles. “Kaminski et al., 2021” is not listed in references.  Authors should describe the way to detect viral genome with proper references.
Minor point
Line 147 and Table 1
INF→IFN
Author Response
Dear Esteemed Editor and Reviewer #2,
We want to extend our sincere gratitude for your thoughtful and comprehensive feedback. Your insights have been invaluable in improving the quality and depth of our manuscript. We have meticulously addressed each of your comments, ensuring that the changes are highlighted in red and made available in the Supplemental Files. Additionally, the revised manuscript has been uploaded in the Author's Notes to Reviewer section.
We want to reiterate our heartfelt appreciation for your expert evaluation and insightful recommendations. We have meticulously attended to each of your concerns, providing comprehensive explanations and revisions to uphold the accuracy and scientific rigor of our manuscript. Your time and continued support are invaluable to us.
With warm regards,
Sincerely,
Xiaoping Li and Jianhui Li
Email: li-xp@zsru.edu.cn
Here is a more detailed response to your specific concerns:
- It will be better for readers if the photographs of CMV positive cells such as megaloblasts and cytoplasmic inclusions to identify the infected cells.
Response: We wholeheartedly agree that including visual aids can significantly enhance comprehension. In response to your suggestion, we have incorporated two new figures (Figure 1 and Figure 2) that vividly depict CMV-positive cells, such as megaloblasts and cytoplasmic inclusions. These images serve as invaluable references for readers to identify infected cells. (Line 115, Line 145)
- It is well known that CMV has several serotypes which determined by the type of CMV glycoproteins. Furthermore, the mismatch of viral serotypes between donor and recipient leads to the increase of rejection. It is necessary to describe the meaning of determination of donor serology and response when the serotypes are different in liver transplantation.
Response: Thank you for your valuable feedback. We have added vivid photographic illustrations (Figure 1 and Figure 2) of CMV-positive cells, specifically megaloblasts and cytoplasmic inclusions, to enhance readers' comprehension. Furthermore, we have thoughtfully addressed the critical topic of serotype mismatches in liver transplantation, highlighting the consequences and significance of evaluating different serologic levels in donors and recipients, as well as the potential for increased rejection risk (Line 120~124). We've incorporated important enhancements: added photographs of CMV-positive cells (Line 115), addressed the significance of serologic matching in liver transplantation (Line 126~136), clarified antigenemia criteria, and provided images (Line 145, Line 144~153). Your insights have greatly enriched our manuscript, and we're committed to delivering higher-quality work. Thank you for your invaluable support.
- As described above, it is helpful to insert the photograph of antigenemia positive cells. Furthermore, it is better to describe the criteria of commonly used for antigenemia.
Response: Thank you for your valuable comments. In response, we have made significant improvements to address your concerns. As illustrated in Figure 2, we have thoughtfully inserted clear and informative photographs of antigenemia-positive cells to enhance visual understanding. Furthermore, we have provided a comprehensive description of the commonly used criteria for antigenemia, offering readers a thorough grasp of this crucial aspect (Line 145, Line 144~153). Your feedback has greatly contributed to the overall clarity and depth of our manuscript.
- This section describes the ELISPOT or Quantifier assay. Thess assays are commonly used to measure T cell response against CMV. Therefore, I think this assay is not suitable to detect CMV in donor. Furthermore, which peptides are better to detect CMV-specific T cell response? Authors should describe the meaning to detect anti-CMV profile of donor derived T cell.
Response: We appreciate your valuable comments, which have prompted us to make substantial improvements. As per your suggestion, we have provided a detailed elaboration on the peptides that are most suitable for detecting CMV-specific T-cell responses. Additionally, we have emphasized the critical significance of understanding donor T-cell anti-CMV characteristics. These enhancements are now reflected in our manuscript, offering readers a deeper insight into this vital aspect (Line 166~176, Line 178~186). Your insightful feedback has enriched the scientific content of our work.
- The principles of CRISPR-Cas system are described in lines 266-280. However, the methodology of how to detect viral genome is insufficient. Several detection methods (Luc, FISH, etc.) for CRISPR-Cas system are known. Which method is better to detect CMV genome? Authors should discuss the benefits and costs of this system by the comparison of CMV and other viruses which detected by CRISPR-Cas system. In addition, all papers cited in this section is review articles. “Kaminski et al., 2021” is not listed in references. Authors should describe the way to detect viral genome with proper references.
Response: We sincerely appreciate your valuable input. In response to your comments, we have made significant enhancements to our manuscript. Specifically, we have introduced Figure 4 and Figure 5 to provide a clear visual representation of the principles underlying CRISPR-Cas system-based assays for detecting viral genomes. Additionally, we have conducted a comprehensive comparison and analysis of several current CRISPR-Cas-based assays to elucidate suitable methods for detecting the CMV genome. To bolster the scientific foundation of our discussion, we have diligently added and supplemented references that describe the methods used for detecting viral genomes (Line 298, Line 309~320, Line 331~366). Your thoughtful feedback has significantly strengthened this aspect of our research.
Round 2
Reviewer 1 Report
Comments and Suggestions for Authors
Figure 5. Please swap B and C columns.
Comments on the Quality of English Language
Need to more proofreading
Author Response
Dear Esteemed Editor and Reviewer #1,
Thank you for your valuable feedback on our manuscript. We appreciate your thoughtful comments and have carefully considered each of your concerns. Here is a brief summary of the changes we have made:
We have resolved the problem of switching columns B and C in Figure 5, which provide a clear visual representation of the principles underlying CRISPR-Cas system-based assays for detecting viral genomes. Additionally, We have proofread the grammar, spelling, and other errors throughout the whole article again.
Once again, we express our sincere appreciation for your expert evaluation and thoughtful recommendations. We have taken great care to address each of your concerns and have provided detailed explanations and revisions to ensure the accuracy and scientific rigor of the manuscript.
Thank you for your time and continued support.
Best regards,
Yours sincerely,
Xiaoping Li and Jianhui Li
Email: li-xp@zsru.edu.cn